# (−)-Lariciresinol Isolated from the Roots of *Isatis indigotica* Fortune ex Lindl. Inhibits Hepatitis B Virus by Regulating Viral Transcription

**DOI:** 10.3390/molecules27103223

**Published:** 2022-05-18

**Authors:** Lu Yang, Huiqiang Wang, Haiyan Yan, Kun Wang, Shuo Wu, Yuhuan Li

**Affiliations:** 1CAMS Key Laboratory of Antiviral Drug Research, Institute of Medicinal Biotechnology, Chinese Academy of Medical Sciences and Peking Union Medical College, Beijing 100050, China; yanglu@imb.pumc.edu.cn (L.Y.); wanghuiqiang@imb.pumc.edu.cn (H.W.); yanhaiyan@imb.pumc.edu.cn (H.Y.); wangkun@imb.pumc.edu.cn (K.W.); 2Beijing Key Laboratory of Antimicrobial Agents, Institute of Medicinal Biotechnology, Chinese Academy of Medical Sciences and Peking Union Medical College, Beijing 100050, China; 3NHC Key Laboratory of Biotechnology of Antibiotics, Institute of Medicinal Biotechnology, Chinese Academy of Medical Sciences and Peking Union Medical College, Beijing 100050, China

**Keywords:** (−)-lariciresinol, HBV, HBV RNA, viral transcription, HNF1α

## Abstract

Chronic hepatitis induced by hepatitis B virus (HBV) infection is a serious public health problem, leading to hepatic cirrhosis and liver cancer. Although the currently approved medications can reliably decrease the virus load and prevent the development of hepatic diseases, they fail to induce durable off-drug control of HBV replication in the majority of patients. The roots of *Isatis indigotica* Fortune ex Lindl., a traditional Chinese medicine, were frequently used for the prevention of viral disease in China. In the present study, (−)-lariciresinol ((−)-LRSL), isolated from the roots of *Isatis indigotica* Fortune ex Lindl., was found to inhibit HBV DNA replication of both wild-type and nucleos(t)ide analogues (NUCs)-resistant strains in vitro. Mechanism studies revealed that (−)-LRSL could block RNA production after treatment, followed by viral proteins, and then viral particles and DNA. Promoter reporter assays and RNA decaying dynamic experiments indicated that (−)-LRSL mediated HBV RNA reduction was mainly due to transcriptional inhibition rather than degradation. Moreover, (−)-LRSL in a dose-dependent manner also inhibited other animal hepadnaviruses, including woodchuck hepatitis virus (WHV) and duck hepatitis B virus (DHBV). Combining the analysis of RNA-seq, we further found that the decrease in HBV transcriptional activity by (−)-LRSL may be related to hepatocyte nuclear factor 1α (HNF1α). Taken together, (−)-LRSL represents a novel chemical entity that inhibits HBV replication by regulating HNF1α mediated HBV transcription, which may provide a new perspective for HBV therapeutics.

## 1. Introduction

Chronic hepatitis B (CHB) is a contagious viral disease commonly seen across the globe. Despite NUCs and interferon alpha (IFNα) having been accepted to manage CHB infection, a curative treatment for the disease is still hard to realize.

HBV is a noncytopathic hepatotropic virus pertaining to the *hepadnaviridae* family. The virion contains a 3.2 kb partial double-stranded relaxed circular DNA (rcDNA) comprising four overlapping open reading frames (ORFs) governed by four promoters (Cp, preS1, preS2, X) and two enhancer regions (EnhⅠ, EnhⅡ). The four ORFs encode four mRNAs with different lengths [1,2]. Specifically, HBV is characterized by species specificity and hepatocyte tropism. After the viral entry, the viral capsid releases rcDNA into the nucleus afterward, converted into a covalently closed circular DNA (cccDNA), which is a transcriptional template of viral RNA. The 0.7 kb transcript serves as the HBV X protein (HBx) template. The 2.1/2.4 kb RNAs are responsible for encoding HBV large surface protein (L), middle surface protein (M), and small surface protein (S). The 3.5 kb pregenomic RNA (pgRNA) encodes HBV polymerase (Pol) and core protein (HBc). In addition, it also serves as a template for reverse transcription to form rcDNA [3].

Natural products derived from plants are sources of new drugs. The roots of *Isatis indigotica* Fortune ex Lindl. belong to the family *Brassicaceae* and are widely distributed in China. It is also known as BanLanGen in Chinese and has been used in the clinical treatment of influenza for hundreds of years in China and other Asian countries [4]. In addition to its anti-influenza activity, the roots of *I. indigotica* also have anti-HBV activity. “Fufang Biejia Ruangan Tablet” and the “QiangGan capsule” are reported to have clinical anti-hepatitis B activity and can alleviate clinical symptoms and liver fibrosis in patients with CHB, while the roots of *I. indigotica* are one of their main components [5,6,7,8,9]. Polysaccharides isolated from this plant and pharmacological studies also showed that some of them exhibited antiviral and hepatocyte protective activities [10,11]. In recent years, lignans have also been isolated from the roots of *I. indigotica* [12]. Previous studies have found that lignans from various natural products have anti-HBV activities, such as helioxanthin (HE-145) isolated from the heartwood of *Taiwania cryptomerioides* Hayata, honokiol from *Streblus asper*, etc. [13,14,15,16]. However, no lignans isolated from the roots of *I. indigotica* have been reported to have anti-HBV activities. In this study, (−)-LRSL was isolated from water-soluble chemical constituents in the roots of *I. indigotica* and showed anti-HBV activity in vitro, and its antiviral mechanism was explored.

HBV transcription is regulated by transcription factors and additional co-factors. Host transcription factors interact with DNA sequences upstream of HBV, e.g., promoter regions, to control the transcriptional activity. These transcription factors can be divided into the following two groups: one is unique to hepatocytes, and the other exists in all cell types [17]. Hepatocyte nuclear factor (HNF) is one of the biggest families of liver-specificity transcription factors, which belongs to the nuclear receptor superfamily. Generally, HNF1α, HNF3, and HNF4α have been reported as positive regulatory factors for HBV transcription [18]. HNF3α/β/γ hetero/homodimers upregulate the transcriptional activity of the preS promoter, EnhI, and EnhII by binding them. HNF4α mainly targets the core promoter and reinforces its transcription in hepatoma cells [19]. HNF1α promotes viral transcription in the following two ways: one is to activate HBV preS1P activity via binding to its enhancer/promoter, and the other is to enhance HBV CP activity via combining the HBV EnhⅡ B2 region [18,20]. Flavonoids (Baicalin) [21], phenolic acids (protocatechuic acid) [22], and terpenoids (Saikosaponin C) [23] have been proved to down-regulate HNF1α and inhibit HBV transcription, indicating that regulation on HNF1α of host hepatocytes is a potential therapeutic approach. However, no lignans have been reported to play an anti-HBV role by regulating HNF1α mediated HBV RNA transcription. This study also provides a reference for the study of the anti-HBV mechanism of lignans.

## 2. Results

### 2.1. (−)-LRSL Has Anti-HBV Activity

Four lignans isolated from the roots of *Isatis indigotica* Fortune ex Lindl. were evaluated for their anti-HBV activity. A cytopathic effect (CPE) assay and qPCR assay were applied to test the cytotoxicity and anti-HBV activity of these four compounds, respectively. Among them, compound 1, (−)-LRSL, showed significant antiviral activity with a 50% effective concentration (EC_50_) of 42.62 μM in HepG2.2.15 cells, and it was thus selected for further studies (Figure 1).

In order to exclude the antiviral activity perhaps caused by the compound-mediated cytotoxicity, the cytotoxicity of (−)-LRSL was determined by CCK-8 analysis. The CC_50_ value of (−)-LRSL was greater than 750 μM, and the cell viability was >90% at 750 μM (Figure 2A). Therefore, all subsequent experiments were carried out under 750 μM. To verify the effect of (−)-LRSL, HepG2.2.15 cells were treated with (−)-LRSL at the indicated concentrations for 6 days, and the culture medium was renewed every three days. In these (−)-LRSL-treated cells, the intracellular HBV core DNA was decreased dose-dependently (Figure 2B).

### 2.2. (−)-LRSL Inhibits HBV Replication Including NUC-Resistant Strains

Long-term therapy of CHB with NUCs can trigger the emergence of drug-resistant HBV strains with specific mutations in the polymerase genes and result in therapeutic failures. We subsequently tested the effects of (−)-LRSL on the wild-type plasmid (pHY536207, genotype B) or NUC-resistant HBV genotype B (pHY634 and pHY6923) and genotype C (pHY6945) clones derived from Chinese sufferers resistant to 3TC or ADV [24]. As anticipated, while pHY634 displayed a remarkable resistance to 3TC and pHY6923 and pHY6945 were resistant to ADV, the level of HBV core DNA in cells subjected to pHY536207, pHY634, pHY6923 or pHY6945 transfection was significantly inhibited in the presence of (−)-LRSL (Figure 3A–C). These results reveal that (−)-LRSL exhibits antiviral activity against HBV, and there is no cross-resistance between (−)-LRSL with 3TC and ADV.

### 2.3. (−)-LRSL Suppressed the Secretion of HBV Antigen and Viral Particles in HepG2.2.15 Cells

A complete HBV replication cycle is completed with the secretion of virus particles. (−)-LRSL was then evaluated for its ability to reduce HBV-related products, including HBV DNA (viral particles), extracellular HBeAg, and HBsAg in the supernatant of HepG2.2.15 cells. As presented in Figure 4A–C, (−)-LRSL demonstrated a robust inhibition of HBV DNA and modest inhibition of HBsAg and HBeAg dose-dependently in HepG2.2.15 cells. Moreover, we studied the efficacy of (−)-LRSL after different days of treatment. (−)-LRSL showed a remarkable inhibitory effect on the extracellular HBV DNA, HBsAg, and HBeAg on the second and third days posttreatment (Figure 4D–F). The outcomes revealed the suppressive effects of (−)-LRSL prolongation with the extension of drug therapy duration. The results above indicate that (−)-LRSL significantly inhibits HBV production in a sustained manner.

### 2.4. (−)-LRSL Treatment Decreases the Production of HBV RNAs

The generation of HBV virions and HBsAg within the culture medium could be influenced by some factors in their biosynthesis pathways, including RNA transcription, particle assembly, and secretion. For the purpose of determining the step(s) in virions and HBsAg generation and secretion pathway susceptible to (−)-LRSL, we initially determined viral RNA levels by Northern blot analysis. HepG2.2.15 cells were treated with (−)-LRSL at 300 μM and 100 μM for 6 days. The cells were also treated with the HBV DNA polymerase inhibitor 3TC and capsid assembly modulators HAP derivative Bay41-4109 as positive controls [25]. Consistent with the proposed mechanism, 3TC and Bay41-4109 failed to alter the level of HBV RNA (Figure 5E). In contrast, (−)-LRSL reduced the amount of HBV RNA, with other indicators also tested in the follow-up. Consistent with their antiviral mechanism, 3TC treatment remarkably reduced the amount of viral DNA (Figure 5D) but failed to change the levels of core protein (Figure 5C), capsid (Figure 5A), and encapsidated pgRNA (Figure 5B). Bay41-4109 treatment thoroughly abrogated protein production, capsid formation, and hence pgRNA encapsidation and DNA synthesis (Figure 5A–D). Differently, (−)-LRSL treatment dose-dependently reduced the levels of all markers, including capsids (Figure 5A), encapsidated pgRNA (Figure 5B), HBV core DNA (Figure 5D), and HBV core protein (Figure 5C). The results thus imply that (−)-LRSL has a unique antiviral mechanism that can inhibit HBV replication, probably via decreasing the production of HBV RNA.

### 2.5. (−)-LRSL Inhibits the Replication of Different Species of Hepadnaviruses

The above results proved that (−)-LRSL can also inhibit the formation and assembly of core proteins and the production of DNA, which is similar to Bay41-4109. As hepadnaviruses from different species have varying degrees of genetic relationship, we examined the antiviral activities of the (−)-LRSL against other members of the hepadnaviridae family, including WHV and DHBV. HepG2 cells were transfected with plasmids of different species of hepadnaviruses regulated by the CMV promoter (pCMV-HBV, pCMV-DHBV, and pCMV-WHV), and viral replication was assessed by qPCR. Consistent with the previous reports, 3TC treatment induced a reduction of HBV core DNA in three hepadnavirus transfected cells (Figure 6C) [26]. The results showed that Bay41-4109 was able to inhibit merely HBV and its closely related WHV but not distantly related DHBV because it is a kind of HBV core protein allosteric modulator (CpAMs) that has specific recognition for HBV core protein (Figure 6B). However, the core DNA level of hepadnaviruses from different sources was significantly reduced in the presence of (−)-LRSL, about 64% for HBV, 57% for DHBV, and 50% for WHV at 300 µM (Figure 6A). This experiment not only identified the antiviral spectrum of (−)-LRSL but also helped to determine its unique antiviral mechanism distinguished from CpAMs.

### 2.6. (−)-LRSL Exerts Antiviral Effects by Inhibiting HBV RNA Transcription Rather than Degradation

The inhibition of HBV RNA can be achieved by speeding up the degradation of HBV RNA or inhibiting the transcription of HBV. In order to determine whether (−)-LRSL mediated down-regulation of HBV RNA was owing to a transcriptional or post-transcriptional mechanism, we directly measured the decaying dynamics of HBV RNA in the presence of (−)-LRSL. Briefly, HepG2.2.15 cells were treated with Act-D to shut down de novo HBV pgRNA transcription. (−)-LRSL and DHQ, a compound that accelerates HBV RNA degradation, were also added in combination with Act-D, respectively, and the decaying dynamics of HBV RNA were determined in a time-course research. As presented in Figure 7A,B, compared with the control group, the velocity of HBV RNA degradation in HepG2.2.15 cells was more rapid in the presence of DHQ, which is in agreement with the previous report [27]. In contrast, (−)-LRSL did not demonstrate a significant effect on the degradation rate of HBV RNA (Figure 7A,B). Hence, we speculated that (−)-LRSL-mediated HBV RNA decrease may be possible through transcriptional inhibition.

To further confirm the above hypothesis, reporter plasmids that carried HBV promoters that regulate the transcription of EnhII/Cp, preS1, and preS2 were constructed [28]. Each DNA fragment was inserted upstream into the luciferase gene. In addition, pGL4.75 plasmid expressing CMV promoter with *Renilla* luciferase was used as control (Figure 7C). (−)-LRSL exerted different inhibitory effects on HBV EnhII/Cp, preS1 and preS2 promoters, of which the effect on EnhII/Cp and preS1 promoters was significant (Figure 7D,E). Taken together, we conclude that (−)-LRSL mediates HBV RNA reduction not through accelerating HBV RNA decay but due to transcriptional inhibition.

### 2.7. (−)-LRSL May Inhibit HBV RNA Transcription by Inhibiting HNF1α

Given the antiviral effect of (−)-LRSL achieved by suppressing HBV transcription, we subsequently embarked on elucidating the underlying mechanism. The RNA-seq-based transcriptome analysis was employed to estimate the transcriptome changes in HepG2.2.15 cells. In total, 2800 differential expression genes, including 1356 up-regulated genes and 1444 down-regulated genes, were selected according to a strict screening threshold (*p*-adjust < 0.001) (Figure 8A). In total, 150 genes were assigned to the Clusters of Orthologous Groups (COG) functional classification “Transcription” (Figure 8B). Further, GO functional enrichment analysis revealed that only six genes regulate DNA-templated transcription, and the enrichment factor was the highest, which represented the highest enrichment degree (Figure 8C,D). Among these six genes, HNF1α has been reported to regulate HBV transcription. Coincidentally, the action site of HNF1α with HBV is consistent with the promoter region regulated by (−)-LRSL.

### 2.8. The Inhibitory Effect of (−)-LRSL on HNF1α Is Synergistic with its Inhibitory Effect on HBV RNA

The reduction of HNF1α mRNA level and protein level was further validated by RT-qPCR and Western blot (Figure 9A,B). As HNF1α always plays a role with other members of HNFs such as HNF3α and HNF4α, the expression of these genes at protein and mRNA levels was also verified. It was found that (−)-LRSL had a significant down-regulation effect on HNF3α but not HNF4α at the mRNA level. To some extent, the expression level of HNF3α also decreased, but the regulation of HNF1α was more obvious than that of HNF3α. Referring to the above, HNF1α was selected for subsequent mechanism verification. To confirm the vital role of HNF1α, a time-course study was performed to explore the suppressive role of (−)-LRSL on HBV RNA, HNF1α, and core protein. It was found that the level of HBV RNA and HNF1α mRNA decreased in the cells treated with (−)-LRSL during the period from 16 h to 32 h (Figure 9C,D). Intriguingly, (−)-LRSL had no significant inhibitory effect on core protein within 32 h. So, achievement of HBV core protein reduction might need long periods of treatment since (−)-LRSL dose-dependently reduced the level of HBV core protein after 6 days of treatment (Figure 5C). These data suggest that (−)-LRSL treatment affects an early stage of HBV replication due to the synthesis of HBV core protein occurring in the posttranscriptional stage.

## 3. Discussion

The roots of *Isatis indigotica* Fortune ex Lindl., as traditional Chinese medicine, have been used in clinical for the treatment of various diseases, including viral infectious diseases. The chemical components isolated from the roots of *I. indigotica*, including alkaloids, flavonoids, etc., exhibit satisfactory performances in the treatment of a variety of diseases. Pharmacological studies of polysaccharides isolated from the roots of *I. indigotica* showed anti-HBV activity in HepG2.2.15 cells [10]. In China, “Fufang Biejia Ruangan Tablet” and “QiangGan capsule” with the roots of *I. indigotica* as the main component have been put into clinical use, which have hepatocyte protective activities in patients with CHB [5,6,7,8,9]. In recent years, lignans have also been isolated from the roots of *I. indigotica*. Among them, clemastanin B and lariciresinol-4-*O*-*β*-D-glucopyranoside have been reported to display anti-influenza virus activity, which proves that lignans isolated from the roots of *I. indigotica* have potential in antiviral activity [29,30]. In our efforts to screen antiviral agents, we found that (−)-LRSL could inhibit both wild-type HBV. (Figure 2) and NUCs-resistant HBV (Figure 3) replication in vitro. Further studies found that (−)-LRSL could also inhibit HBV DNA (viral particles) in the supernatant, secreted HBsAg and HBeAg (Figure 4). In the mechanistic study, it was found that (−)-LRSL significantly inhibited the levels of intracellular HBV core DNA, total HBV RNA, encapsidated pgRNA, core protein, and capsids in multiple HBV replication stages in a dose-dependent manner, which indicated that its antiviral mechanism was different from positive drugs (3TC and Bay41-4109), and the results revealed its antiviral action by inhibiting HBV RNA (Figure 5). The observed significant inhibitory effect of (−)-LRSL on DHBV and WHV also supported its broad-spectrum antiviral activity (Figure 6). Moreover, (−)-LRSL reduced HBV RNA by transcription inhibition rather than RNA degradation, in a different way from DHQ, a compound that can accelerate HBV RNA degradation resulting in HBV RNA reduction and then affect HBsAg expression [27] (Figure 7). Based on these results and in combination with layer upon layer analysis of RNA-seq, we found that HNF1α might mediate HBV RNA transcription inhibition induced by (−)-LRSL (Figure 8). This is consistent with the inhibitory effect of (−)-LRSL on HBV promoters.

HNFs are a class of liver-specific transcription factors that regulate liver gene expression. They are pivotal for hepatocyte differentiation and hepatic metabolism. HNF1α, which belongs to the HNF family, is crucial for HBV replication [31]. In addition to HNF1α, HNF family members include HNF3 and HNF4, which are also essential for HBV replication by regulating transcription [32,33]. In fact, there is some controversy regarding the role of HNF1α in regulating the replication of HBV. Previous studies have revealed that HNF1α can interact with other HNF1 family proteins to form homodimers/heterodimers and increase the transcriptional activity of preS1 facilitated by the POU domain [32,34]. Additionally, HNF1α interacts with preS1, the core promoter, and the EnhII element, which is crucial for HBV replication [35,36]. However, an in vivo study suggested that the absence of HNF1α did not prevent HBV replication [37]. Even when a high level of HNF1α activates NF-κB, HNF1α can also inhibit HBV transcription [38].

In summary, our data demonstrate that (−)-LRSL inhibits HBV transcription at least partially by suppressing HNF1α. Our findings expand the antiviral spectrum of lignans isolated from the roots of *I. indigotica* and suggest that there may be other unknown mechanisms in the regulation of HBV transcription. Further studies are warranted to elucidate how (−)-LRSL regulates this process in more detail.

## 4. Materials and Methods

### 4.1. Compounds

(−)-LRSL (98.6% HPLC purity), (–)-lariciresinol-4,4′-bis-*O*-*β*-D-glucopyranoside (98.5% HPLC purity), (–)-lariciresinol-4-*O*-*β*-D-glucopyranoside (98.8% HPLC purity) and (–)- lariciresinol-4′-*O*-*β*-D-glucopyranoside (98.9% HPLC purity) isolated from the roots of *Isatis indigotica* Fortune ex Lindl. were provided by Professor Jiangong Shi, Department of Chemosynthesis, Institute of Materia Medica, Chinese Academy of Medical Sciences, and Peking Union Medical College (Beijing, China). The structure information of all the tested compounds was listed in the Appendix A, which is the same as that reported [30,39,40,41]. The stock solution concentration of these four compounds is 300 mM in DMSO and the chemical structures were shown in Figure 1. Lamivudine (3TC, MedChemExpress, Monmouth Junction, NJ, USA), Adefovir (ADV, MedChemExpress), Bay41-4109 (MedChemExpress), Actinomycin D (Act-D, MedChemExpress), and Dihydroquinolizinone (DHQ, synthesized in-house) were dissolved in DMSO as stock solution (1 mM) and stored at −20 °C. These compounds were diluted with cell culture media before use.

### 4.2. Cells

Human hepatoma cells HepG2 were maintained in Minimum Essential Medium (MEM, Gibco, Waltham, MA, USA) added with 10% fetal bovine serum (FBS, Gibco), 100 U/mL penicillin, and 100 μg/mL streptomycin (Gibco). HepG2.2.15 cells that involved complete HBV genome and were able to realize steady HBV replication were maintained in MEM added with 10% FBS, 100 U/mL penicillin, 100 μg/mL streptomycin, and 400 μg/mL G418 (Gibco) as the culture medium.

### 4.3. Plasmids

pCMV-HBV, pCMV-DHBV and pCMV-WHV expressing viral pregenomic RNA (pgRNA) controlled by cytomegalovirus (CMV) immediate early promoter and wild-type HBV replicon pHBV1.3mer were kindly provided by Professor Ju-Tao Guo (Baruch S. Blumberg Institute, Doyletown, PA, USA) [26]. The data with regard to the sequence and construction of the wild-type or NUC-resistant HBV genotype B and C plasmids, such as pHY536207, pHY634, pHY6923, and pHY6945, have been described in the past [42].

### 4.4. Cytotoxicity Assay

The cytotoxicity of (−)-lariciresinol in HepG2.2.15 was analyzed by Cell Counting Kit-8 (CCK-8) (TransGen Biotech, Beijing, China) assay. The absorbance was determined at 450 nm with EnSpire multimode plate reader (PerkinElmer, Waltham, MA, USA).

### 4.5. Transient-Transfection Assay

HepG2 cells seeded in 12-well dishes were transfected with indicated plasmid (0.5 µg) using Lipofectamine 3000 (Invitrogen, Carlsbad, CA, USA). In total, 6 h after transfection, the culture media was substituted with fresh media containing various concentrations of the compounds and cultured for extra 72 h. The intracellular HBV core DNA was tested via the experiments as follows.

### 4.6. Southern Blot

The HBV replicative intermediate DNA was extracted as previously described [43]. The HBV DNA was dissolved in 1.5% agarose gel and transferred to Hybond-XL membrane (GE Healthcare, Buckinghamshire, UK) via blotting for the hybridization with a DIG-labelled minus-strand-specific HBV probe synthesized by DIG HBV RNA labeling mix (Roche, Basel, Switzerland) through an in vitro transcription kit (Promega, Fitchburg, WI, USA). The hybridization signal was detected with Chemidoc XRS+ chemiluminescence imaging analysis system (Bio-Rad, Hercules, CA, USA). The densitometric signal from Southern blot was subjected to quantification via Image Lab program.

### 4.7. Quantitative PCR (qPCR)

HBV capsid-related DNA in the culture medium was subjected to quantification based on a standard curve generated from the standards with known nucleic acid quantities. In short, supernatants acquired from the 24-well dishes were clarified according to the method provided by the HBV nucleic acid quantitative determination kit (Sansure Biotech Inc, Changsha, Hunan, China), and the HBV DNA quantification was completed with Applied Biosystems 7500 Fast Real-Time PCR System instrument (Applied Biosystems, Foster City, CA, USA) for detection.

Intracellular HBV core DNA was detected by qPCR with TransStart Green qPCR SuperMix (TransGen Biotech). The reaction condition was as follows: 94 °C for 30 s, 94 °C for 5 s, 60 °C for 30 s, and 40 cycles. The DNA expressing level was calculated by the relative threshold cycle (ΔΔCT) method.

### 4.8. Northern Blot

The HBV total RNA of the cells was isolated using TRIzol (Invitrogen) and dissolved in DEPC water [44]. The viral pgRNA was quantified by a Northern blot assay, and RNA was denatured and transferred to a formaldehyde agarose gel as depicted in the past [43]. After UV detected quantitative 18 S and 28 S rRNA, RNA was blotted onto Hybond-XL membrane (GE), and UV cross-linked. The membrane was hybridized in DIG specifically labeled HBV DNA probe. Hybridization signal was detected with Chemidoc XRS+ chemiluminescence imaging analysis system (Bio-Rad).

### 4.9. Quantitative Reverse Transcription PCR (RT-qPCR)

HBV pgRNA, HBV encapsidated pgRNA, HNF1α, HNF3α, and HNF4α mRNAs in HepG2.2.15 cells were analyzed by RT-qPCR normalized to the level of the internal control gene, β-actin. The PCR analysis was run with HiScript II One-Step RT-qPCR SYBR Green Kit (Vazyme Biotech, Nanjing, Jiangsu, China) and Applied Biosystems 7500 Fast Dx Real-Time qPCR system according to the following thermocycling parameters below: 50 °C for 3 min, followed by 95 °C for 30 s and 40 cycles of 95 °C for 10 s and 60 °C for 30 s. Sequences of the primers used are available upon request.

### 4.10. HBsAg and HBeAg Quantification

The supernatant was collected, and the expressing level of hepatitis B e antigen (HBeAg) and hepatitis B surface antigen (HBsAg) in cellular supernatants were evaluated using an HBeAg/HBsAg detection kit (Chemclin, Shanghai, China). The luminescence value was measured on the EnVision multilabel microplate reader (PerkinElmer).

### 4.11. Western Blot Analysis

Western blotting was carried out according to a method depicted in the past [45]. The cells were lysed in the lysis buffer (2% SDS, 5% β-mercaptoethanol, 60 mM Tris-HCl, pH6.8, 0.01% bromphenol blue, and 10% glycerol). The signal was detected using Omni-ECL Femto Light Chemiluminescence Kit (EpiZyme, Shanghai, China). The primary antibodies used in the experiment include β-actin, HNF1α, HNF3α, HNF4α (Cell Signaling Technology, Danvers, MA, USA), and HBc (Synthesized by GenScript, Nanjing, Jiangsu, China).

### 4.12. Construction of HBV Promoter Reporter Plasmid

For the purpose of constructing an HBV Cp reporter plasmid, a fragment that covers the HBV EnhII and Cp region (nt 1400–1820) was amplified by PCR from plasmid pHBV1.3 and introduced to the HindIII and SacI restriction sites in pGL4.17-Basic vector (Promega). The plasmid was designated EnhII/Cp-Luc, in which expression of the firefly luciferase reporter gene is governed by the HBV core promoter. HBV surface promoter-reporter plasmids, namely, preS1-Luc and preS2-Luc, were engineered through insertion of HBV DNA fragments encompassing preS1 (nt 2708–2809) and preS2 (nt 2850–3180) promoter region into the HindIII and SacI restriction sites in pGL4.17-Basic, separately. CMV promoter *Renilla* luciferase reporter plasmid pGL4.75 was purchased from Promega.

### 4.13. Statistical Analysis

Statistical significance was assessed using one-way analysis of Variance (ANOVA) with Dunnett’s test with a threshold of * *p* < 0.05, ** *p* < 0.01, and *** *p* < 0.001.

## Figures and Tables

**Figure 1 molecules-27-03223-f001:**
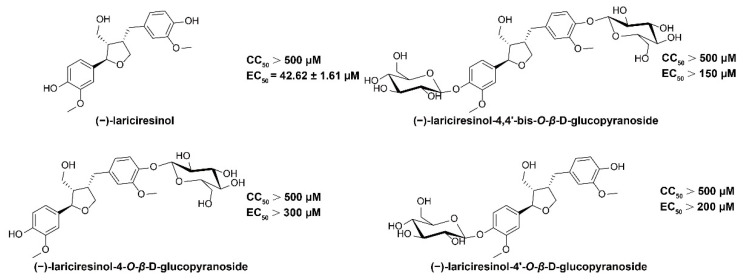
The chemical structures of four compounds isolated from the roots of *Isatis indigotica* Fortune ex Lindl.

**Figure 2 molecules-27-03223-f002:**
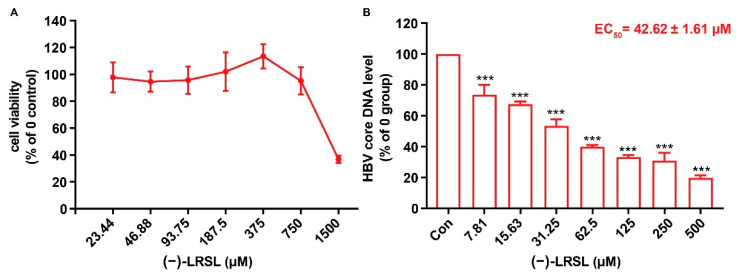
(−)-LRSL has anti-HBV activity. (**A**) HepG2.2.15 cells were treated with indicated concentrations of (−)-LRSL for 3 days, cell viability was determined using the CCK-8 assay. (**B**) HepG2.2.15 cells were treated with indicated concentrations of (−)-LRSL for 6 days, the level of intracellular HBV core DNA was identified by qPCR. *** *p* < 0.001.

**Figure 3 molecules-27-03223-f003:**
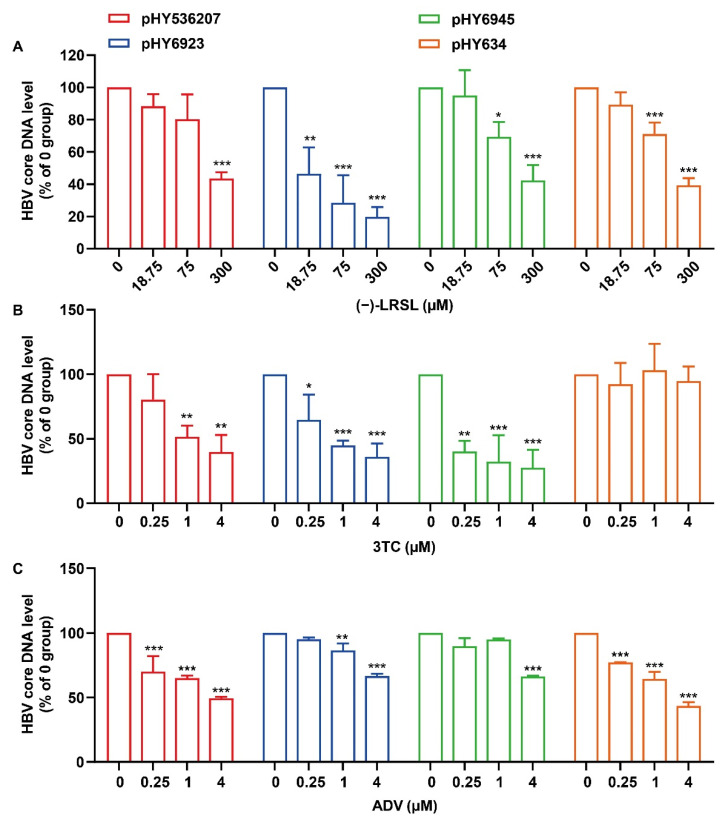
Inhibition of (−)-LRSL on HBV replication including NUC-resistant strains. HepG2 cells were transfected with pHY536207, pHY634, pHY6923, and pHY6945, followed by treatment with compound (**A**) (−)-LRSL (300, 75, 18.75 µM), (**B**) 3TC (4, 1, 0.25 µM), and (**C**) ADV (4, 1, 0.25 µM) for 3 days, then the content of intracellular HBV DNA was detected by real-time PCR. Data were expressed as the average ± SD of three independent experiments. * *p* < 0.05, ** *p* < 0.01, and *** *p* < 0.001.

**Figure 4 molecules-27-03223-f004:**
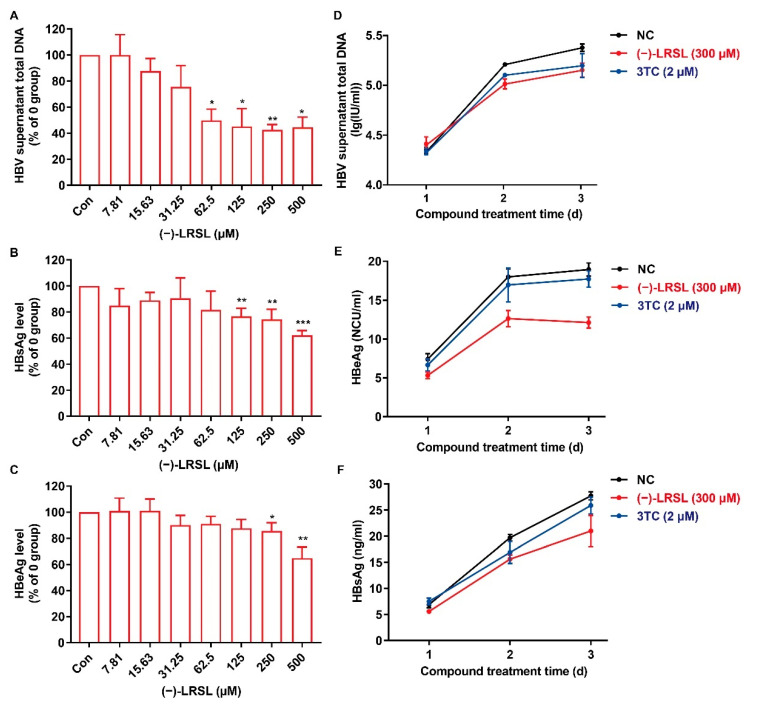
(−)-LRSL suppressed the secretion of viral antigen and viral particles. HepG2.2.15 cells were treated with indicated concentrations of (−)-LRSL for 6 days. (**A**) The level of HBV DNA (viral particles) within the supernatant was detected by qPCR. (**B**,**C**) HBsAg and HBeAg in supernatant were measured by ELISA assay. HepG2.2.15 cells were treated with 3TC (2 µM) or indicated concentrations of (−)-LRSL. (**D**) The level of HBV DNA (viral particles) within the supernatant was detected by qPCR. (**E**,**F**) HBeAg and HBsAg in supernatant was measured by ELISA assay. * *p* < 0.05, ** *p* < 0.01, and *** *p* < 0.001.

**Figure 5 molecules-27-03223-f005:**
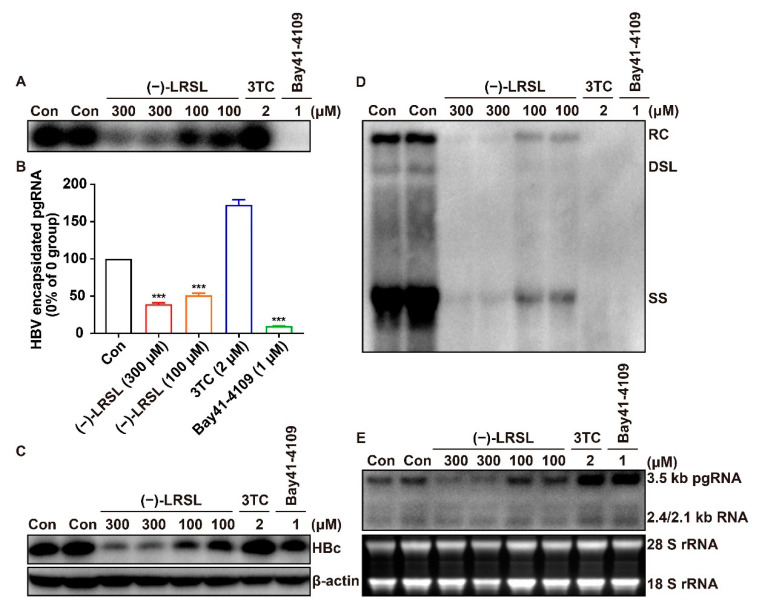
HepG2.2.15 cells were treated, or mock treated with the indicated concentrations of the compound (−)-LRSL (between 150 and 300 µM), 3TC (2 µM), Bay41-4109 (1 µM) for 6 days. (**A**) The total amounts of capsids were determined by a particle gel assay in a 1.5% agarose gel electrophoresis. (**B**) Encapsidated pgRNA was extracted and determined by qRT-qPCR. (**C**) HBc expression was detected by Western blotting with a rabbit polyclonal antibody, and β-actin served as a loading control. (**D**) HBV DNA replicative intermediates were extracted and determined by Southern blot. A DIG-labeled full-length minus-strand-specific riboprobe was used for Southern blot analysis. RC, relaxed circular DNA; DSL, double-stranded linear DNA; SS, single-strand, negative-polarity DNA. (**E**) Intracellular viral RNA was identified by Northern blotting hybridization. In addition, 28 S and 18 S rRNA were utilized as loading controls. *** *p* < 0.001.

**Figure 6 molecules-27-03223-f006:**
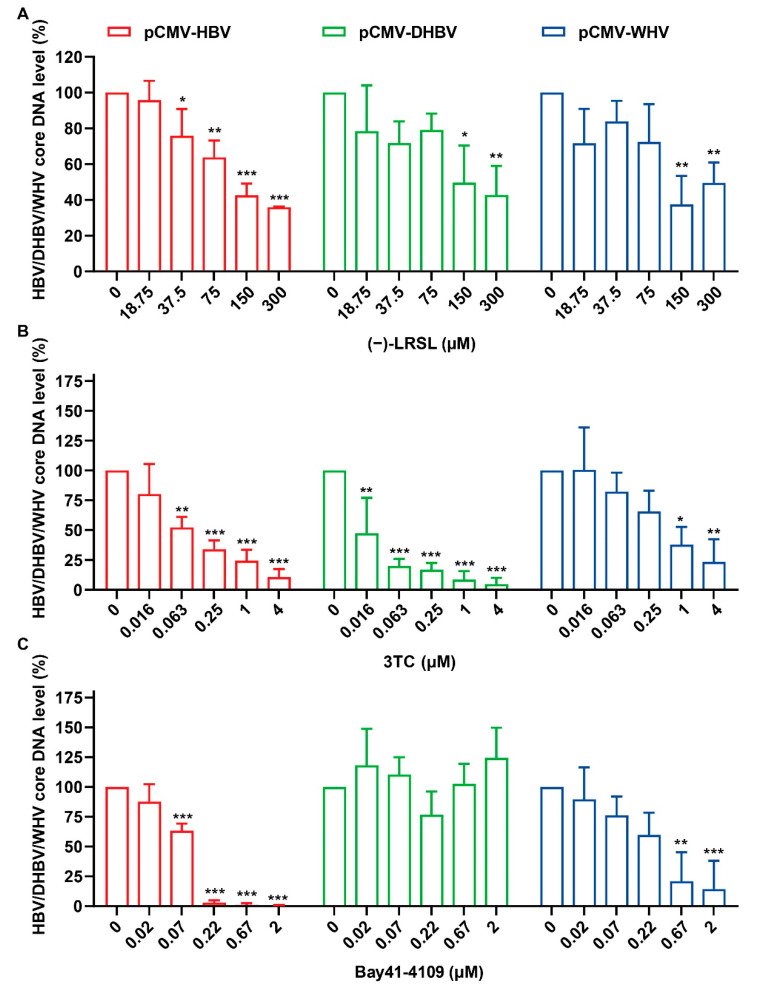
Inhibition of (−)-LRSL on different species hepadnaviruses. HepG2 cells were transfected with pCMV-HBV, pCMV-DHBV, and pCMV-WHV, followed by the treatment with compound (**A**) (−)-LRSL (300, 150, 75, 37.5, 18.75 µM), (**B**) 3TC (4, 1, 0.25, 0.063, 0.016 µM), and (**C**) Bay41-4109 (2, 0.67, 0.22, 0.07, 0.02 µM) for 3 days, then the content of intracellular HBV DNA was detected by real-time PCR. * *p* < 0.05, ** *p* < 0.01, and *** *p* < 0.001.

**Figure 7 molecules-27-03223-f007:**
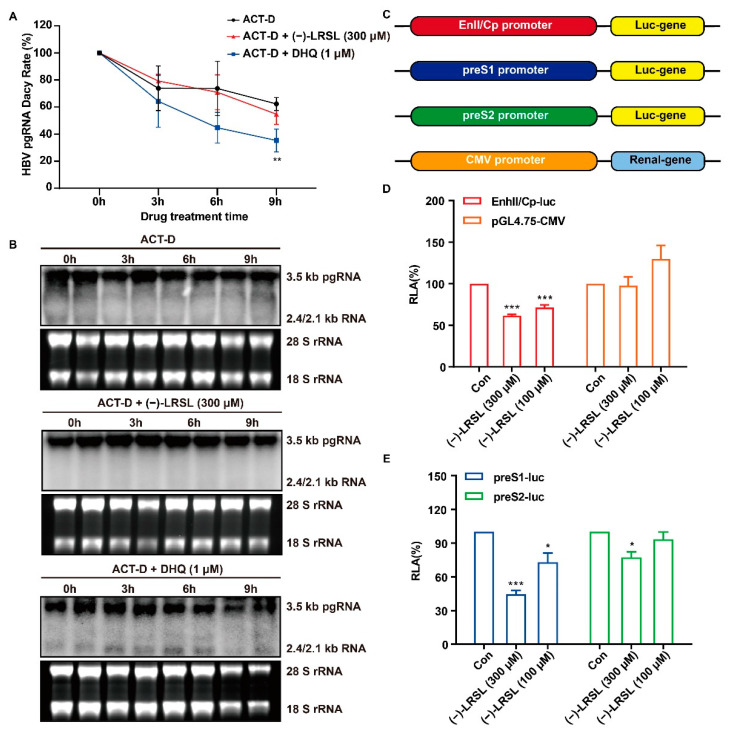
Effects of (−)-LRSL on HBV RNA transcription and degradation. (**A**,**B**) (−)-LRSL lost its inhibitory activity when de novo RNA synthesis was terminated. HepG2.2.15 cells were cultured in 6-well dishes and treated with (−)-LRSL at 300 μM and Act-D at 10 μg/mL for indicated time length. Total cellular RNA was extracted, then qRT-qPCR and Northern blot were performed to detect the change of HBV mRNA at different time after drug treatment. (**C**) Diagram of reporter plasmids carrying HBV promoters. (**D**,**E**) (−)-LRSL treatment fails to change HBV core promoter activity in transfected cells. HepG2 cells were seeded in a 96-well plate and co-transfected with 100 ng of PreS1-Luc, PreS2-Luc, EnhII/Cp-Luc and 10 ng of pGL4.75-CMV, plus (−)-LRSL (300 μM). Two days posterior to the transfection, cells were harvested, and luciferase activities were measured. The plotted relative luciferase activity (RLA) represents the mean standard deviation (SD, n = 3) of the ratios of absorbance obtained from wells treated with (−)-LRSL acquired from the wells that were transfected with control vector. * *p* < 0.05, ** *p* < 0.01, and *** *p* < 0.001.

**Figure 8 molecules-27-03223-f008:**
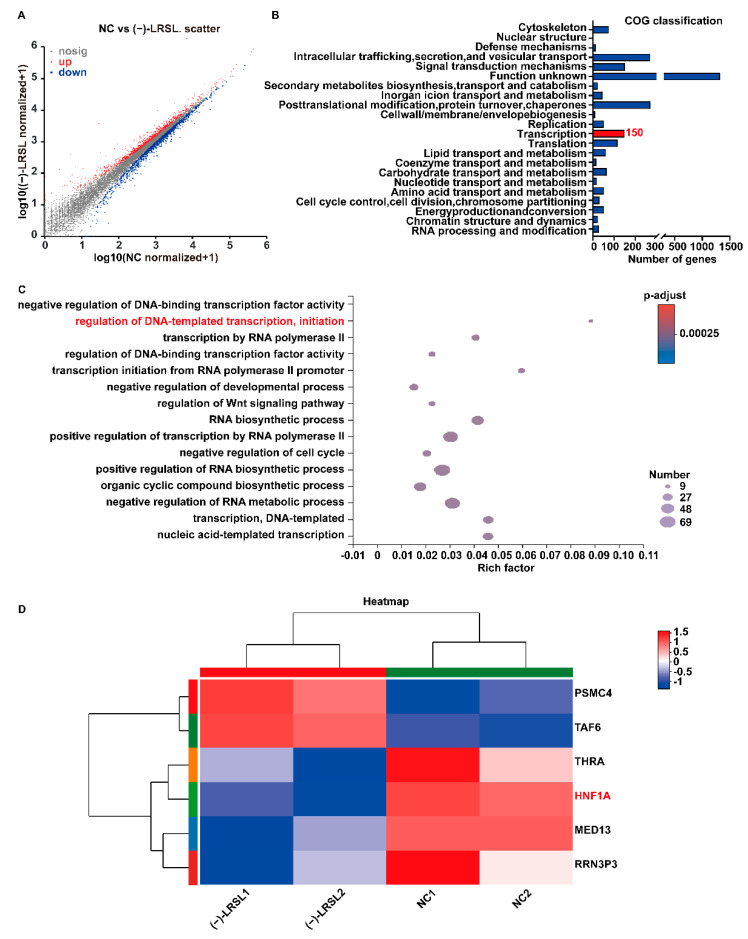
HNF1α level was significantly downregulated after (−)-LRSL treatment. (**A**) Representative Scatter Plot of 2800 significant genes (1356 upregulated genes marked in red and 1444 downregulated genes marked in blue) for NC vs. (−)-LRSL treatment (*p*-adjust < 0.001) in HepG2.2.15 cells. (**B**) COG classification analysis based on the RNA-seq results for NC vs. (−)-LRSL treatment. (**C**) GO enrichment analysis based on the genes regulating transcription. (**D**) A heatmap shows cluster analysis of 6 genes (marked in red in panel (**C**)).

**Figure 9 molecules-27-03223-f009:**
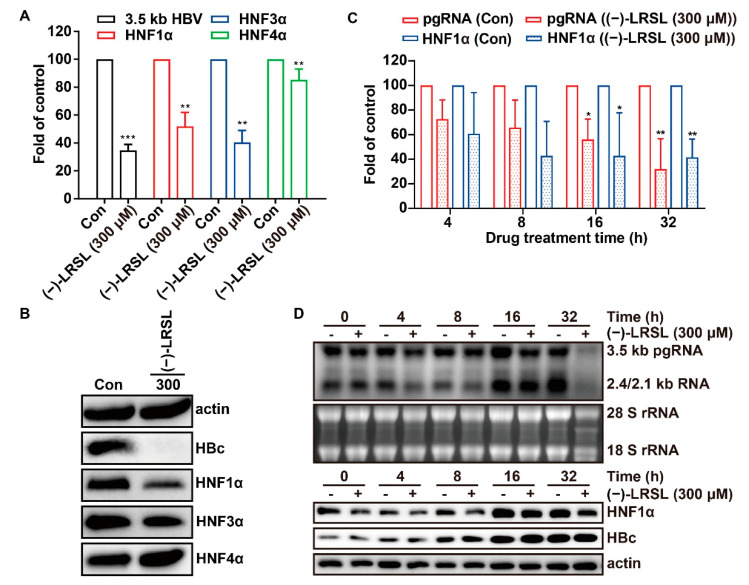
The inhibitory effect of (−)-LRSL on HNF1α is synergistic with its inhibitory effect on HBV RNA. HepG2.2.15 cells were cultured mock treated or treated with (−)-LRSL (300 µM) for 6 days, and mRNA of HNFs was determined by RT-qPCR (**A**), while protein of HNFs was determined by Western blot (**B**). HepG2.2.15 cells were treated with (−)-LRSL (300 µM) for indicated hours, HBV RNA and HNF1α mRNA was determined by RT-qPCR (**C**) and Northern blot (**D**). Protein of HNFs and HBc was detected by Western blot (**D**). * *p* < 0.05, ** *p* < 0.01 and *** *p* < 0.001.

## Data Availability

The original contributions presented in the study are included in the article. RNA-seq data were analyzed on the free online platform of Majorbio Cloud Platform The datasets presented in this study can be found in online repositories. The names of the repository and accession number can be found below: NCBI, PRJNA785716.

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
