# Peer review of "(−)-Lariciresinol Isolated from the Roots of Isatis indigotica Fortune ex Lindl. Inhibits Hepatitis B Virus by Regulating Viral Transcription"

_molecules, 2022, doi:10.3390/molecules27103223_

Round 1
Reviewer 1 Report
In view of the growing interests in natural or plant-derived anti-HBV molecules, this a very good research which has delineated the detailed mechanism of antiviral activity of the isolated compound at molecular level. With minor grammatical and spelling corrections, the revised manuscript can be accepted.

Author Response
Response to Reviewer 1 Comments
Point : In view of the growing interests in natural or plant-derived anti-HBV molecules, this a very good research which has delineated the detailed mechanism of antiviral activity of the isolated compound at molecular level. With minor grammatical and spelling corrections, the revised manuscript can be accepted.
Response : Thanks for your suggestion. According to the comments, we have modified the manuscript and the revised content can be viewed in the resubmitted manuscript. The modification details are as follows:
- page 1, lines 29-31: “Moreover, (−)-LRSL dose dependently inhibit other animal hepadnaviruses, including woodchuck hepatitis virus (WHV) and duck hepatitis B virus (DHBV).” has been modified to” Moreover, (-)-LRSL in a dose-dependent manner also inhibited other animal hepadnaviruses, including woodchuck hepatitis virus (WHV) and duck hepatitis B virus (DHBV).”
- page 6, lines 180: “species hepadnaviruse” has been modified to”species of hepadnaviruses”.
- page 8, lines 207: “posttranscriptional” has been modified to”post-transcriptional”.
- page 8, lines 224: “we came to the conclusion that” has been modified to”we conclude that”.
- Page 12, lines 306: “The result that (−)-LRSL has significant inhibitory effect on” has been modified to“The observed significant inhibitory effect of (−)-LRSL on”.
- Page 12, lines 334-336: “compund 2” has been modified to”compund”,then “compund 3” and “compund 4” has been deleted.
Reviewer 2 Report
The present work is focused on Isatis indigotica roots, a traditional Chinese medicine, frequently used in China for the prevention of viral diseases. Specifically, in the present study, (-)-lariciresinol ((-)-LRSL), isolated from such aroot, was found to inhibit HBV DNA replication of both wild-type and nucleos(t)ide analogues (NUC)-resistant strains in vitro.
The work is interesting and expand the anti-viral spectrum of lignans ((-)-LRSL) suggesting that there may be other unknown mechanisms in the regulation of HBV transcription by HNF1α. Surely, further studies are warranted to elucidate the way how (-)-LRSL regulate this process in a more “comprehensive” way.
I do have some remarks prior to be considered for publication.
- English needs a moderate revision e.g. lines 19-20, the sentence should be re-phrased.
- Introduction should be enlarged by inclusing a more detailed literature survey. For instance as the authors reported in lines 303-307, other natural molecues (not lignans but a flavone and a phenolic acid) e.g. baicalein and protocatechuic acid, were also proved to exert an anti-HBV effect by inhibiting HNF1α or HNF1α related pathways. The novelty of the work needs to be more emphasized.
- Section 4.1. Compounds. It is not clear how these compounds were isolated. What was the concentration of stock solutions beforer after dilution with cell culture media?
Author Response
Response to Reviewer 2 Comments
The present work is focused on Isatis indigotica roots, a traditional Chinese medicine, frequently used in China for the prevention of viral diseases. Specifically, in the present study, (-)-lariciresinol ((-)-LRSL), isolated from such aroot, was found to inhibit HBV DNA replication of both wild-type and nucleos(t)ide analogues (NUC)-resistant strains in vitro.
The work is interesting and expand the anti-viral spectrum of lignans ((-)-LRSL) suggesting that there may be other unknown mechanisms in the regulation of HBV transcription by HNF1α. Surely, further studies are warranted to elucidate the way how (-)-LRSL regulate this process in a more “comprehensive” way.
I do have some remarks prior to be considered for publication.
Point 1: English needs a moderate revision e.g. lines 19-20, the sentence should be re-phrased..
Response 1: Thanks for your suggestion. The manuscript has been modified moderately. And the sentense “Although current standard care can reliably decrease the virus load and prevent the development of hepatic diseases, they still can't cure these diseases” of lines 19-20 ” has been modified to “Although the current approved medications can reliably decrease the virus load and prevent the development of hepatic diseases, they fail to induce a durable off-drug control of HBV replication in the majority of patients”.
Point 2: Introduction should be enlarged by inclusing a more detailed literature survey. For instance as the authors reported in lines 303-307, other natural molecues (not lignans but a flavone and a phenolic acid) e.g. baicalein and protocatechuic acid, were also proved to exert an anti-HBV effect by inhibiting HNF1α or HNF1α related pathways. The novelty of the work needs to be more emphasized.
Response 2: Thanks for this invaluable comment. According to your suggestion, we modifed the theoretical background in the introduction part and the revised content can be viewed in the resubmitted manuscript (line 79-86). In the resubmitted manuscript, we introduced the existing flavonoids, phenolic acids and terpenoids have been proved to down-regulate HNF1α and inhibit HBV transcription. Moreover, it is emphasized that no lignans have been reported to play an anti-HBV role through similar mechanisms. Besides, we enlarged the theoretical background in the introduction part including the related research of the roots of Isatis indigotica Fortune on anti-HBV (line 54-67, ref 5-16 were added), which can better provide a theoretical basis for our research.
Point 3: Section 4.1. Compounds. It is not clear how these compounds were isolated.
Response 3: Thank you very much for your question. The separation flow is shown in the following flowchart (can be seen in an attached word file). The relevant results of this part are still being sorted out. We have determined the structures of all the tested compounds by spectroscopic data analysis, chemical method, and theoretical calculation. We list the structure information of (−)-LRSL, which was also the aglycone of compounds 2−4.
The structure information of (−)-LRSL is as follows: White amorphous powder, [α]20 D –21.5 (c 0.87, CH3OH); CD (CH3OH): 216 (Δε –0.17), 235 (Δε +0.79), 283 (Δε +0.24) nm; 1H NMR (CD3OD, 500 MHz) δ 6.85 (1H, d, J = 1.5 Hz, H-2), 6.71 (2H, m, H-5å’ŒH-6), 4.69 (1H, d, J = 7.0 Hz, H-7), 2.32 (1H, m, H-8), 3.78 (1H, overlap, H2-9a), 3.57 (1H, dd, J = 11.0, 3.5 Hz, H2-9b), 6.74 (1H, d, J = 1.5 Hz, H-2¢), 6.65 (1H, dd, J = 8.0 Hz, H-5¢), 6.58 (1H, dd, J = 8.0, 1.5 Hz, H-6¢), 2.87 (1H, dd, J = 13.5, 4.5 Hz, H2-7¢a), 2.43 (1H, dd, J = 11.0, 13.5 Hz, H2-7¢b), 2.67 (1H, m, H-8′), 3.92 (1H, dd, J = 8.0, 6.0 Hz, H2-9¢a), 3.67 (1H, dd, J = 8.0, 4.0 Hz, H2-9¢b), 3.78 (3H, s, OCH3-3), 3.77 (3H, s, OCH3-3¢); 13C NMR (CD3OD, 125 MHz) δ 137.5 (C-1), 110.5 (C-2), 149.0 (C-3), 147.0 (C-4), 116.0 (C-5), 119.8 (C-6), 84.0 (C-7), 54.1 (C-8), 60.4 (C-9), 133.5 (C-1¢), 113.3 (C-2¢), 149.0 (C-3¢), 145.8 (C-4¢), 116.2 (C-5¢), 122.2 (C-6¢), 33.6 (C-7¢), 43.9 (C-8¢), 73.5 (C-9¢), 56.3 (OCH3-3å’ŒOCH3-3¢); (+)-HR-ESI-MS at m/z 383.1462 [M + Na]+ (C20H24O6Na, 383.1465).
Point 4: What was the concentration of stock solutions beforer after dilution with cell culture media?
Response 4: Thank you very much for your question. The concentration of stock solutions of (−)-LRSL before dilution with cell culture media is 300mM. Relevant contents have been added in the revised manuscript (line 340).

Reviewer 3 Report
Dear Authors,
Please use the correct plan name, Isatis tinctoria L. [Brasicaceae], see https://powo.science.kew.org/
Where is the rationale behind the study? why you suppose that the plant or compound are effective against HBV? References 8 and 9 does not provide any evidence.
Please provide a reference to consult the correct structure elucidation of the tested compound.
Methods: “6 hours after transfection, the culture media was substituted with fresh media containing various concentrations of the compounds and cultured for extra 72h.”
Which concentrations? Where are the control groups?
“There is a long history of medicinal application of natural products stemming from plants. Isatis indigotica roots, as a traditional Chinese medicine, is famous all over the world for its broad-spectrum antiviral activity.” Yes, but against influenza virus, again where is the rationale behind this study?
Author Response
Response to Reviewer 3 Comments
Point 1: Please use the correct plan name, Isatis tinctoria L. [Brasicaceae], see https://powo.science.kew.org/.
Response 1: Thank you very much for your question. We confirmed our plant name according to the website you provided and published papers. The plant name is “Isatis indigotica Fortune” distributed in China and the dried roots of Isatis indigotica Fortune was selected for isolation. Its name is “the roots of Isatis indigotica Fortune” (can be seen in an attached word file), also called “BanLanGen” in the pharmacopoeia of the People's Republic of China (ChP). The team providing this compound has published several articles named after it [1, 2].
Point 2: Where is the rationale behind the study? why you suppose that the plant or compound are effective against HBV? References 8 and 9 does not provide any evidence.
Response 2: Thanks for this invaluable comment. According to your suggestion, we modifed the theoretical background of this study and the revised content can be viewed in the resubmitted manuscript (line 54-67). Because the roots of Isatis indigotica Fortune has anti-HBV activity (ref 5-11 were added), and lignans from other plants are also reported to have anti-HBV activity (ref 13-16 was added), so we supposed that lignans from the roots of Isatis indigotica Fortune may also exhibit an antiviral activity against HBV. In the subsequent screening experiment, we did found that (−)-LRSL inhibit HBV DNA replication.
Point 3: Please provide a reference to consult the correct structure elucidation of the tested compound.
Response 3: Thank you very much for your question. The separation flow is shown in the following flowchart (can be seen in an attached word file). The relevant results of this part are still being sorted out. We have determined the structures of all the tested compounds by spectroscopic data analysis, chemical method, and theoretical calculation. We list the structure information of (−)-LRSL, which was also the aglycone of compounds 2−4.
The structure information of (−)-LRSL is as follows: White amorphous powder, [α]20 D –21.5 (c 0.87, CH3OH); CD (CH3OH): 216 (Δε –0.17), 235 (Δε +0.79), 283 (Δε +0.24) nm; 1H NMR (CD3OD, 500 MHz) δ 6.85 (1H, d, J = 1.5 Hz, H-2), 6.71 (2H, m, H-5å’ŒH-6), 4.69 (1H, d, J = 7.0 Hz, H-7), 2.32 (1H, m, H-8), 3.78 (1H, overlap, H2-9a), 3.57 (1H, dd, J = 11.0, 3.5 Hz, H2-9b), 6.74 (1H, d, J = 1.5 Hz, H-2¢), 6.65 (1H, dd, J = 8.0 Hz, H-5¢), 6.58 (1H, dd, J = 8.0, 1.5 Hz, H-6¢), 2.87 (1H, dd, J = 13.5, 4.5 Hz, H2-7¢a), 2.43 (1H, dd, J = 11.0, 13.5 Hz, H2-7¢b), 2.67 (1H, m, H-8′), 3.92 (1H, dd, J = 8.0, 6.0 Hz, H2-9¢a), 3.67 (1H, dd, J = 8.0, 4.0 Hz, H2-9¢b), 3.78 (3H, s, OCH3-3), 3.77 (3H, s, OCH3-3¢); 13C NMR (CD3OD, 125 MHz) δ 137.5 (C-1), 110.5 (C-2), 149.0 (C-3), 147.0 (C-4), 116.0 (C-5), 119.8 (C-6), 84.0 (C-7), 54.1 (C-8), 60.4 (C-9), 133.5 (C-1¢), 113.3 (C-2¢), 149.0 (C-3¢), 145.8 (C-4¢), 116.2 (C-5¢), 122.2 (C-6¢), 33.6 (C-7¢), 43.9 (C-8¢), 73.5 (C-9¢), 56.3 (OCH3-3å’ŒOCH3-3¢); (+)-HR-ESI-MS at m/z 383.1462 [M + Na]+ (C20H24O6Na, 383.1465).
Point 4: Methods: “6 hours after transfection, the culture media was substituted with fresh media containing various concentrations of the compounds and cultured for extra 72h.” Which concentrations? Where are the control groups?
Response 4: We did not mention the specific concentrations of compounds in the method, because we transfected different plasmidsin different experiments, and the concentration of positive control compounds and (−)-LRSL were also different. Therefore, we have add a detailed description in the figure legends of figures 3 and 6.
Point 5: “There is a long history of medicinal application of natural products stemming from plants. Isatis indigotica roots, as a traditional Chinese medicine, is famous all over the world for its broad-spectrum antiviral activity.” Yes, but against influenza virus, again where is the rationale behind this study?
Response 5: Thanks for this invaluable comment. According to your suggestion, we modifed the discussion part and the revised content can be viewed in the resubmitted manuscript (line 54-67 and line 290-294). In the resubmitted manuscript, besides influenza virus, we enlarged the theoretical background in the introduction part including the related research of the roots of Isatis indigotica Fortune on anti-HBV (ref 5-10 were added), which can better provide a theoretical basis for our research.
References:
- Meng, L.; Guo, Q.; Liu, Y.; Chen, M.; Li, Y.; Jiang, J.; Shi, J., Indole alkaloid sulfonic acids from an aqueous extract of Isatis indigotica roots and their antiviral activity. Acta Pharm Sin B 2017, 7, (3), 334-341.
- Meng, L.; Guo, Q.; Liu, Y.; Shi, J., 8,4'-Oxyneolignane glucosides from an aqueous extract of "ban lan gen" (Isatis indigotica root) and their absolute configurations. Acta Pharm Sin B 2017, 7, (6), 638-646.

Round 2
Reviewer 3 Report
Dear Authors,
Minor, the correct name of the plant is Isatis indigotica Fortune ex Lindl. Please mention it at least once in the text.
Major
The structure information of (−)-LRSL is as follows.., this information must be included in the manuscript together with the hplc profiles of the other compounds, It can be added as supplementary material.
The identity of the compounds must be clear in the manuscript, at this point is not clear what do you test.
Thanks
Author Response
Response to Reviewer 3 Comments
Point 1: the correct name of the plant is Isatis indigotica Fortune ex Lindl. Please mention it at least once in the text.
Response 1: Thanks for this invaluable comment. According to your suggestion, we modifed the name of the plant as Isatis indigotica Fortune ex Lindl. and the revised content can be viewed in the resubmitted manuscript.
Point 2: The structure information of (−)-LRSL is as follows.., this information must be included in the manuscript together with the hplc profiles of the other compounds, It can be added as supplementary material. The identity of the compounds must be clear in the manuscript, at this point is not clear what do you test.
Response 2: Thanks for your invaluable comment. Although the main body of this article is biological research, we still modified the description of the compounds in the revised manuscript (line 336-344) and Figue 1 according to your suggestion. We also supplemented the structural information of (−)-LRSL and the other compounds in the supplementary materials.